# NORMALIZATION GRADIENTS ARE LEAST-SQUARES RESIDUALS

## ABSTRACT

Batch Normalization (BN) and its variants have seen widespread adoption in the deep learning community because they improve the training of deep neural networks. Discussions of why this normalization works so well remain unsettled. We make explicit the relationship between ordinary least squares and partial derivatives computed when back-propagating through BN. We recast the back-propagation of BN as a least squares fit, which zero-centers and decorrelates partial derivatives from normalized activations. This view, which we term *gradient-least-squares*, is an extensible and arithmetically accurate description of BN. To further explore this perspective, we motivate, interpret, and evaluate two adjustments to BN.

## 1 INTRODUCTION

Training deep neural networks has become central to many machine learning tasks in computer vision, speech, and many other application areas. Ioffe & Szegedy (2015) showed empirically that Batch Normalization (BN) enables deep networks to attain faster convergence and lower loss. Reasons for the effectiveness of BN remain an open question (Lipton & Steinhardt, 2018). Existing work towards explaining this have focused on covariate shift; Santurkar et al. (2018) described how BN makes the loss function smoother. This work examines the details of the back-propagation of BN, and recasts it as a least squares fit. This gradient regression zero-centers and decorrelates partial derivatives from the normalized activations; it passes on a scaled residual during back-propagation. Our view provides novel insight into the effectiveness of BN and several existing alternative normalization approaches in the literature.

### 1.1 CONTRIBUTIONS

Foremost, we draw an unexpected connection between least squares and the gradient computation of BN. This motivates a novel view that complements earlier investigations into why BN is so effective. Our view is consistent with recent empirical surprises regarding ordering of layers within ResNet residual maps (He et al., 2016b) and within shake-shake regularization branches (Huang & Narayanan, 2018). Finally, to demonstrate the extensibility of our view, we motivate and evaluate two variants of BN from the perspective of *gradient-least-squares*. In the first variant, a least squares explanation motivates the serial chaining of BN and Layer Normalization (LN) (Ba et al., 2016). In the second variant, regularization of the least-squares leads to a version of BN that performs better on batch size two. In both variants, we provide empirical support on CIFAR-10.

In summary, our work presents a view, which we term *gradient-least-squares*, through which the back-propagation of BN and related work in a neural network can be recast as least squares regression. This regression decomposes gradients into an explained portion and a residual portion; BN back-propagation will be shown to remove the explained portion. Hopefully, *gradient-least-squares* will be broadly useful in the future design and understanding of neural network components. Figure 1 reviews normalization with batch statistics, and illustrates our main theorem.

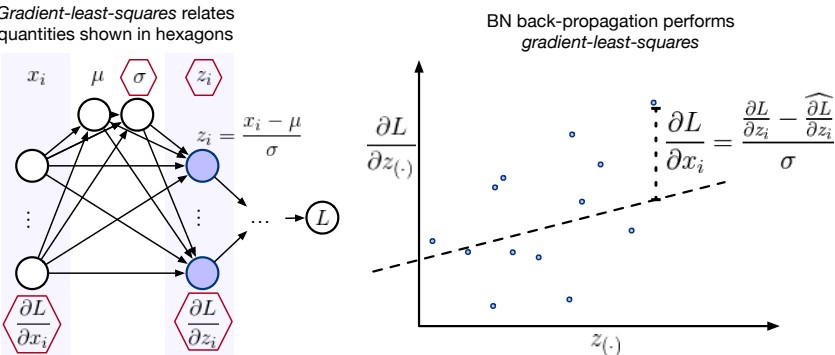

Figure 1: The left figure reviews, for a single channel at a particular layer within a single batch, notable quantities computed during the forward pass and during back-propagation of BN. Let $\{x_i\}_{i=1...N}$ be activations. Let $\mu = \sum_{i=1}^{N} \frac{x_i}{N}$ and $\sigma^2 = \sum_{i=1}^{N} \frac{(x_i - \mu)^2}{N}$. Let $L$ be a function dependent on the normalized activations $z_i$ defined for each $j$ by $z_j = \frac{(x_j - \mu)}{\sigma}$ This, along with partial derivatives, are shown in the left figure. Our work establishes a novel identity on the quantities shown in hexagons. The right figure illustrates our main result in a scatter plot, in which each pair $\left( z_i, \frac{\partial L}{\partial z_i} \right)$ is shown as a data point in the regression.

## 2 NORMALIZATION GRADIENTS ARE LEAST-SQUARES RESIDUALS

Consider any particular channel within which $\{x_i\}$ are activations to be normalized in BN moment calculations. Ioffe & Szegedy (2015) defined BN as

$$BN(\boldsymbol{x}) = \frac{(\boldsymbol{x} - \mu)}{\sigma} \cdot c + b \qquad (1)$$

where $\sigma, \mu$ are batch moments, but $b$ and $c$ are learned per-channel parameters persistent across batches. In BN, the batch dimension and spatial dimensions are marginalized out in the computation of batch moments. For clarity, we consider a simplified version of BN. We ignore the variables $b$ and $c$ in equation 1 responsible for a downstream channel-wise affine transformation. Ignoring $b$ and $c$ is done without loss of generality, since the main observation in this work will focus on the Gaussian normalization and remains agnostic to downstream computations. We also ignore a numerical stability hyperparameter $\epsilon$.

We examine back-propagation of partial derivatives through this normalization, where $\mu$ and $\sigma$ are viewed as functions of $\boldsymbol{x}$. Notably, $\mu$ and $\sigma$ are functions of each $x_i$, and thus the division by $\sigma$ is not affine. We write the normalized output as

$$\boldsymbol{z} = \frac{(\boldsymbol{x} - \mu)}{\sigma} \qquad (2)$$

We review ordinary least squares of a single variable with intercept (Friedman et al., 2001).

Let $g_j = \alpha + \beta z_j + \epsilon_j$ where $\alpha$ and $\beta$ are parameters, $\boldsymbol{z}$ and $\boldsymbol{g}$ are observations. $z_j$ and $g_j$ are entries in $\boldsymbol{z}$ and $\boldsymbol{g}$ respectively. $\epsilon_j$ are i.i.d. Gaussian residuals. We wish to fit $\alpha$ and $\beta$

$$\hat{\alpha}, \hat{\beta} = \arg\min_{\alpha, \beta} \mathbb{E}_j(\|\boldsymbol{g} - \alpha - \beta \boldsymbol{z}\|^2) \qquad (3)$$

The least-squares problem in equation 3 is satisfied by $\hat{\beta} = \frac{\text{Cov}(\boldsymbol{z}, \boldsymbol{g})}{\text{Var}(\boldsymbol{z})}$ and $\hat{\alpha} = \mathbb{E}(\boldsymbol{g}) - \hat{\beta}\mathbb{E}(\boldsymbol{z})$

When $\boldsymbol{z}$ are normalized activations and $\boldsymbol{g}$ are partial derivatives, then $\mathbb{E}\boldsymbol{z} = 0$ and $\mathrm{Var}(\boldsymbol{z}) = 1$. In this special case, the solution simplifies into

$$\hat{\beta} = \mathrm{Cov}(\boldsymbol{z}, \boldsymbol{g}) \tag{4}$$

$$\hat{\alpha} = \mathbb{E}(\boldsymbol{g}) \tag{5}$$

**Theorem 1** (Normalization gradients are least-squares residuals). *Let $i \in \{1 \dots N\}$ be indices over some set of activations $\{x_i\}$. Then the moment statistics are defined by $\mu = \sum_{i=1}^{N} \dfrac{x_i}{N}$ and $\sigma^2 = \sum_{i=1}^{N} \dfrac{(x_i - \mu)^2}{N}$. Let L be a function dependent on the normalized activations $z_i$ defined for each j by $z_j = \dfrac{(x_j - \mu)}{\sigma}$. Then, the gradients of L satisfy, for all $j \in \{1, \dots, N\}$, the following:*

$$\sigma \frac{\partial L}{\partial x_j} = \frac{\partial L}{\partial z_j} - \widehat{\frac{\partial L}{\partial z_j}} \tag{6}$$

*where*

$$\widehat{\frac{\partial L}{\partial z_j}} = \hat{\alpha} + \hat{\beta} z_j \tag{7}$$

$$\hat{\alpha}, \hat{\beta} = \arg\min_{\alpha, \beta} \sum_{i=1}^{N} \left( \frac{\partial L}{\partial z_i} - \alpha - \beta z_i \right)^2 \tag{8}$$

*Proof: Normalization gradients are least-squares residuals.* The proof involves a derivation of partial derivatives by repeated applications of the chain rule and rules of total derivative. Because $\{z_i\}$ normalized over $i$ has mean 0 and variance 1, the partial derivatives can be rearranged to satisfy the single variable ordinary least squares framework.

Fix $j$. We expand $\dfrac{\partial L}{\partial x_j}$ as a linear combination of $\left\{ \dfrac{\partial L}{\partial z_i} \right\}_{i=1\dots N}$

$$\frac{\partial L}{\partial x_j} = \frac{\partial L}{\partial z_j} \frac{\partial z_j}{\partial x_j} + \sum_{i \neq j}^{N} \frac{\partial L}{\partial z_i} \frac{\partial z_i}{\partial x_j} \tag{9}$$

We state $\dfrac{\partial z_i}{\partial x_j}$ directly. Steps are in Appendix A under Lemma 1.

$$\frac{\partial z_i}{\partial x_j} = \begin{cases} \dfrac{-1 - z_j z_i}{\sigma N} & \text{if } i \neq j \\[3mm] \dfrac{N - 1 - z_j^2}{\sigma N} & \text{if } i = j \end{cases} \tag{10}$$

Through substitution of equations 10 into 9, we get

$$\frac{\partial L}{\partial x_j} = \frac{\partial L}{\partial z_j} \frac{N - 1 - z_j^2}{\sigma N} + \sum_{i \neq j}^{N} \left[ \frac{\partial L}{\partial z_i} \cdot \frac{-1 - z_j z_i}{\sigma N} \right] \tag{11}$$

$$\sigma \frac{\partial L}{\partial x_j} = \frac{\partial L}{\partial z_j} + \frac{1}{N} \sum_{i=1}^{N} \left[ (-1 - z_i z_j) \frac{\partial L}{\partial z_i} \right] \tag{12}$$

$$\sigma \frac{\partial L}{\partial x_j} = \frac{\partial L}{\partial z_j} - \left( \frac{1}{N} \sum_{i=1}^{N} \frac{\partial L}{\partial z_i} \right) - \frac{z_j}{N} \sum_{i=1}^{N} z_i \frac{\partial L}{\partial z_i} \tag{13}$$

Noting that $\{z_i\}$ normalized over $i$ has mean 0 and variance 1, we recover $\hat{\beta}$ and $\hat{\alpha}$, in the sense of equations 4 and 5, from equation 13.

$$\frac{1}{N}\sum_{i=1}^{N} z_i \frac{\partial L}{\partial z_i} = \text{Cov}_i\left(z_i, \frac{\partial L}{\partial z_i}\right) = \hat{\beta} \tag{14}$$

$$\frac{1}{N}\sum_{i=1}^{N} \frac{\partial L}{\partial z_i} = \mathbb{E}_i\left[\frac{\partial L}{\partial z_i}\right] - \hat{\beta}\cdot 0 = \hat{\alpha} \tag{15}$$

Finally, we rearrange equations 15 and 14 into 13 to conclude, as desired,

$$\sigma\frac{\partial L}{\partial x_j} = \frac{\partial L}{\partial z_j} - \hat{\alpha} - \hat{\beta}z_j = \frac{\partial L}{\partial z_j} - \widehat{\frac{\partial L}{\partial z_j}} \tag{16}$$

$\square$

During back-propagation of a single batch, the normalization function takes in partial derivatives $\dfrac{\partial L}{\partial z_{(\cdot)}}$, and removes that which can be explained by least squares of $\dfrac{\partial L}{\partial z_{(\cdot)}}$ against $z_{(\cdot)}$. As illustrated in Figure 1, during back-propagation, the residual then divides away $\sigma$ to become $\dfrac{\partial L}{\partial x_{(\cdot)}}$, the gradient for the unnormalized activations.

## 3   RELATED DEEP LEARNING COMPONENTS VIEWED AS GRADIENT CALCULATIONS

BN aims to control its output to have mean near 0 and variance near 1, normalized over the dataset; this is related to the original explanation termed *internal covariate shift* (Ioffe & Szegedy, 2015). Most existing work that improve or re-purpose BN have focused on describing the distribution of activations.

**Definition 1.** *In the context of normalization layers inside a neural network, activations are split into partitions, within which means and variances are computed. We refer to these partitions as* **normalization partitions***.*

**Definition 2.** *Within the context of a normalization partition, we refer to the moments calculated on the partitions as* **partition statistics***.*

Theorem 1 shows that BN has least squares fitting built into the gradient computation. Gradients of the activations being normalized in each batch moment calculation are fit with a single-variable with-intercept least squares model, and only a rescaled residual is kept during back-propagation. We emphasize that the data on which the regression is trained and applied is a subset of empirical activations within a batch, corresponding to the normalization partitions of BN.

To show extensibility, we recast several popular normalization techniques into the *gradient-least-squares* view. We refer to activations arising from a single member of a particular batch as an item. $BHWC$ refers to dimensions corresponding to items, height, width, and channels respectively. In non-image applications or fully connected layers, $H$ and $W$ are 1. BN marginalizes out the items and spatial dimensions, but statistics for each channel are kept separate.

In the subsequent sections, we revisit several normalization methods from the perspective of the gradient. Figure 2 reviews the normalization partitions of these methods, and places our main theorem about *gradient-least-squares* into context.

### 3.1   LAYER NORMALIZATION, INSTANCE NORMALIZATION, GROUP NORMALIZATION

Ba et al. (2016) introduced Layer Normalization (LN) in the context of large LSTM models and recurrent networks. Only the $(H, W, C)$ dimensions are marginalized in LN, whereas BN marginalizes out the $(B, H, W)$ dimensions. In our regression framework, the distinction can be understood

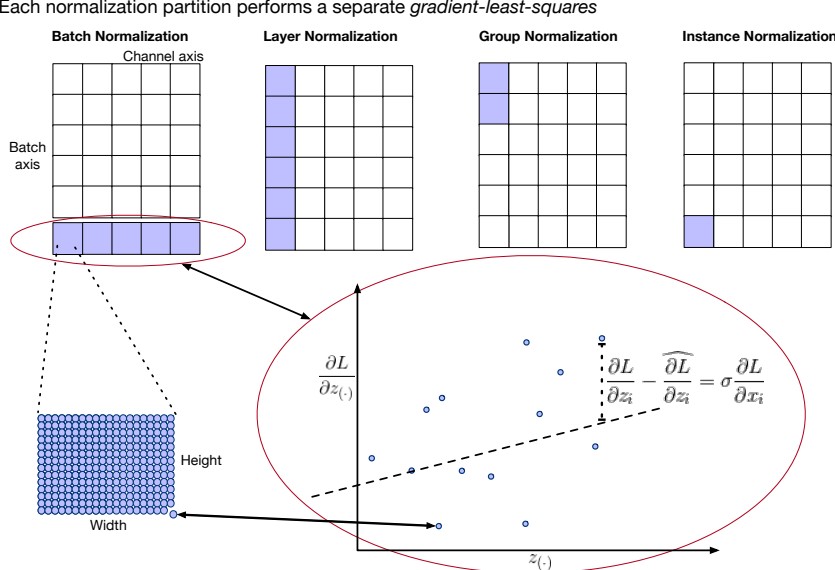

Figure 2: We review the normalization partitions of BN, LN, GN, and IN. Each normalization partition contains a separate set of data points on which the gradient regression is performed. One partition for each method is illustrated in blue. This figure also shows the correspondence between a single activation and a gradient regression data point for BN.

as changing the data point partitions in which least squares are fit during back-propagation. LN marginalizes out the channels, but computes separate statistics for each batch item. To summarize, the regression setup in the back-propagation of LN is performed against other channels, rather than against other batch items.

Huang & Belongie (2017) introduced Instance Normalization (IN) in the context of transferring styles across images. IN is is closely related to contrast normalization, an older technique used in image processing. IN emphasizes end-to-end training with derivatives passing through the moments. Only the $(H, W)$ dimensions are marginalized in IN, whereas BN marginalizes $(B, H, W)$ dimensions. In our framework, this can be understood as using fewer data points and a finer binning to fit the least squares during back-propagation, as each batch item now falls into its own normalization partition.

Wu & He (2018) introduced Group Normalization (GN) to improve performance on image-related tasks when memory constrains the batch size. Similar to LN, GN also marginalizes out the $(H, W, C)$ dimensions in the moment computations. The partitions of GN are finer: the channels are *grouped* into disjoint sub-partitions, and the moments are computed for each sub-partition. When the number of groups is one, GN reduces to LN.

In future normalization methods that involve normalizing with respect to different normalization partitions; such methods can pattern match with BN, LN, IN, or GN; the back-propagation can be formulated as a least-squares fit, in which the partial derivatives at normalized activations $\dfrac{\partial L}{\partial z_{(\cdot)}}$ are fitted against the normalized $z_{(\cdot)}$, and then the residual of the fit is rescaled to become $\dfrac{\partial L}{\partial x_{(\cdot)}}$.

Figure 2 summarize the normalization partitions for BN, LN, IN, and GN; the figure visualizes, as an example, a one-to-one correspondence between an activation in BN, and a data point in the gradient regression.

Theorem 1 is agnostic to the precise nature of how activations are partitioned before being normalized; thus, equation 9 applies directly to any method that partitions activations and performs Gaussian normalization on each partition. The partitioning of BN, LN, IN, and GN are performed

in different respective manners, and each partition is individually subject to Gaussian normalization. Thus, the gradients of BN, LN, IN, and GN are residuals of regressions in the sense of Theorem 1.

## 3.2 WEIGHT NORMALIZATION

Salimans & Kingma (2016) introduced Weight Normalization (WN) in LSTMs, and noted improvements in the condition number of deep networks; WN divides each weight tensor by their respective vector 2-norms. In the view of *gradient-least-squares*, WN has a single-variable intercept-0 regression interpretation in back-propagation, analogous to BN. A raw weight vector $\boldsymbol{v}$, is normalized and scaled before being used as coefficient weights $\boldsymbol{w} = \dfrac{c}{\|\boldsymbol{v}\|}\boldsymbol{v}$, where $c$ is a learned downstream linear scaling parameter.

In this regression setup, the length normalized weights of WN are analogous to the Gaussian normalized activations in BN. We write that $\boldsymbol{z} = \dfrac{\boldsymbol{v}}{\|\boldsymbol{v}\|} = \dfrac{\boldsymbol{w}}{c}$, and state directly an analogous relationship between each $\|\boldsymbol{v}\|\dfrac{\partial L}{\partial v_j}$ and the regression on $\left\{\left(z_i, \dfrac{\partial L}{\partial z_i}\right)\right\}_{i=1\ldots N}$. See Appendix B Lemma 2 for steps that derive the following identity: for loss $L$ and for each component $j$, we have

$$\|\boldsymbol{v}\|\frac{\partial L}{\partial v_j} = \frac{\partial L}{\partial z_j} - \hat{\beta} z_j \tag{17}$$

where

$$\hat{\beta} = \arg\min_{\beta} \|\nabla_{\boldsymbol{z}} L - \beta \boldsymbol{z}\|^2 = (\nabla_{\boldsymbol{z}} L)^T \boldsymbol{z} \tag{18}$$

The L2 normalization of weights in WN appears distinct from the Gaussian normalization of activations in BN; nevertheless, WN can also be recast as a least squares regression.

## 3.3 IDENTITY MAPPINGS IN RESNET, AND SHAKE-SHAKE RESNEXT REGULARIZATION

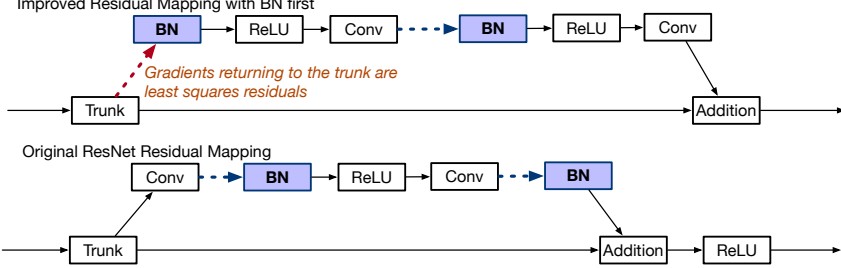

Figure 3: This figure illustrates the original (He et al., 2016a) and improved (He et al., 2016b) residual mappings in ResNets. Arrows point in the direction of the forward pass. Dotted lines indicate that gradients are zero-centered and decorrelated with respect to downstream activations in the residual mapping. The improved ordering has BN coming first, and thus constrains that gradients of the residual map must be decorrelated with respect to some normalized activations inside the residual mapping.

An update to the popular ResNet architecture showed that the network's residual mappings can be dramatically improved with a new ordering (He et al., 2016b). The improvement moved BN operations into early positions and surprised the authors; we support the change from the perspective of *gradient-least-squares*. Figure 3 reviews the precise ordering in the two versions. Huang & Narayanan (2018) provides independent empirical support for the BN-early order, in shake-shake regularization (Gastaldi, 2017) architectures. We believe that the surprise arises from a perspective that views BN only as a way to control the distribution of activations; one would place BN after a sequence of convolution layers. In the *gradient-least-squares* perspective, the first layer of each residual mapping is also the final calculation for these gradients before they are added back into

Table 1: BN plus LN final validation performance (ResNet-34-v2, batch size 128)

| Normalization | CIFAR-10 Accuracy | CIFAR-10 Cross Entropy |
|---|---|---|
| **BN, LN** | **0.9259** | **0.3087** |
| LN, BN | 0.9245 | 0.3389 |
| BN (Ioffe & Szegedy, 2015) | 0.9209 | 0.3969 |
| LN (Ba et al., 2016) | 0.9102 | 0.3548 |

the main trunk. The improved residual branch constrains the gradients returning from the residual mappings to be zero-centered and decorrelated with respect to some activations inside the branch. We illustrate this idea in Figure 3.

## 4 NORMALIZATION APPROACHES MOTIVATED BY LEAST SQUARES

*Gradient-least-squares* views back-propagation in deep neural networks as a solution to a regression problem. Thus, formulations and ideas from a regression perspective would motivate improvements and alternatives to BN. We pursue and evaluate two of these ideas.

### 4.1 BN AND LN AS TWO-STEP GRADIENT REGRESSION

BN and LN are similar to each other, but they normalize over different partitioning of the activations; in back-propagation, the regressions occur respectively with respect to different partitions of the activations. Suppose that a BN and a LN layer are chained serially in either order. This results in a two-step regression during back-propagation; in reversed order, the residual from the first regression is further explained by a second regression on a different partitioning. In principle, whether this helps would depend on the empirical characteristics of the gradients encountered during training. The second regression could further decorrelate partial gradients from activations. Empirically, we show improvement in a reference ResNet-34-v2 implementation on CIFAR-10 relative to BN with batch size 128. In all cases, only a single per-channel downstream affine transformation is applied, after both normalization layers, for consistency in the number of parameters. See table 1 for CIFAR-10 validation performances. We kept all default hyperparameters from the reference implementation: learning schedules, batch sizes, and optimizer settings.

### 4.2 ADDRESSING SMALL BATCHES WITH LEAST-SQUARES REGULARIZATION

BN performs less well on small batches (Ioffe, 2017). *Gradient-least-squares* interprets this as gradient regressions failing on correlated data, an issue typically addressed by regularization. We pursue this idea to recover some performance on small batches by use of regularization. Our regularization uses streaming estimates of past gradients to create virtual data in the regression. This performed better than standard BN on the same batch size, but we did not recover the performance of large batches; this is consistent with the idea that regularization could not in general compensate for having much less data. See Appendix C for CIFAR-10 validation performances.

## 5 LIMITATIONS AND RELATED WORK

### 5.1 SWITCH NORMALIZATION

Luo et al. (2018a) introduced Switch Normalization (SwN), a hybrid strategy for combining moment calculations from LN, BN, and IN. SwN uses learnable scalar logits $\lambda_k$ for $k \in \Omega = \{BN, IN, LN\}$ with corresponding softmax weighting activations $w_k = \frac{\exp(\lambda_k)}{\sum_{k'} \exp(\lambda_{k'})}$ to rescale the contributions to the batch mean for each normalization scheme. It uses an analogous set of parameters $\lambda'_k$ and activations $w'_k$ for variances. We sketch the back-propagation of a simplified version of SN in the perspective of *gradient-least-squares*. We ignore both the division $\epsilon$ and downstream affine

$z \rightarrow c \cdot z + b$. The normalization calculation inside SwN can be written as:

$$z_{bhwc} = \frac{x_{bhwc} - \sum_{k \in \Omega} w_k \mu_{bhwc,k}}{\sqrt{\sum_{k \in \Omega} w'_k \sigma^2_{bhwc,k}}} \tag{19}$$

where $\Omega = \{BN, LN, IN\}$. There is potentially a unique mean and variance used for each activation. Equation 19 bears similarities to the setup in Theorem 1, but we leave unresolved whether there is a *gradient-least-squares* regression interpretation for SN.

## 5.2 DECORRELATED BATCH NORMALIZATION AND SPECTRAL NORMALIZATION

Decorrelated Batch Normalization (DBN) (Huang et al., 2018) is a generalization of BN that performs Mahalanobis ZCA whitening to decorrelate the channels, using differentiable operations. On some level, the matrix gradient equation resemble the least squares formulation in Theorem 1.

Spectral Normalization (SpN) (Miyato et al., 2018) is an approximate spectral generalization of WN. For DBN and SpN, the regression interpretations remain unresolved.

## 5.3 RELATED WORK

BN has been instrumental in the training of deeper networks (Ioffe & Szegedy, 2015). Subsequent work resulted in Batch Renormalization (Ioffe, 2017), and further emphasized the importance of passing gradients through the minibatch moments, instead of a gradient-free exponential running average. In *gradient-least-squares*, use of running accumulators in the training forward pass would stop the gradients from flowing through them during training, and there would be no least-squares. He et al. (2016b) demonstrate empirically the unexpected advantages of placing BN early in residual mappings of ResNet.

Santurkar et al. (2018) showed that BN makes the loss landscape smoother, and gradients more predictable across stochastic gradient descent steps. Balduzzi et al. (2017) found evidence that spatial correlation of gradients explains why ResNet outperforms earlier designs of deep neural networks. Kohler et al. (2018) proved that BN accelerates convergence on least squares loss, but did not consider back-propagation of BN as a least squares residual. Luo et al. (2018b) has recast BN as a stochastic process, resulting in a novel treatment of regularization.

## 6 DISCUSSION, AND FUTURE WORK

This work makes explicit how BN back-propagation regresses partial derivatives against the normalized activations and keeps the residual. This view, in conjunction with the empirical success of BN, suggests an interpretation of BN as a gradient regression calculation. BN and its variants decorrelate and zero-center the gradients with respect to the normalized activations. Subjectively, this can be viewed as removing systematic errors from the gradients. Our view also support empirical results in literature preferring early BN placement within neural network branches.

Leveraging *gradient-least-squares* considerations, we ran two sets of normalization experiments, applicable to large batch and small batch settings. Placing a LN layer either before or after BN can be viewed as two-step regression that better explains the residual. We show empirically on CIFAR-10 that BN and LN together are better than either individually. In a second set of experiments, we address BN's performance degradation with small batch size. We regularize the gradient regression with streaming gradient statistics, which empirically recovers some performance on CIFAR-10 relative to basic BN, on batch size two.

*Why* do empirical improvements in neural networks with BN keep the *gradient-least-squares* residuals and drop the explained portion? We propose two open approaches for investigating this in future work. A first approach focuses on how changes to the gradient regression result in different formulations; the two empirical experiments in our work contribute to this. A second approach examines the empirical relationships between gradients of activations evaluated on the same parameter values; we can search for a shared noisy component arising from gradients in the same normalization partition. Suppose that the gradient noise correlates with the activations – this is plausible because

the population of internal activations arise from using shared weights – then normalizations could be viewed as a layer that removes systematic noise during back-propagation.

In conclusion, we have presented a novel view that reorganizes the back-propagation of BN as a least squares residual calculation. This view generates novel descriptions of normalization techniques related to BN, and comments on the ordering of layers inside the residual mappings of ResNet. This view is extensible and will motivate novel designs of neural network components in future work.

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

## A  PARTIAL DERIVATIVES OF THE NORMALIZATION FUNCTION

**Lemma 1.** *Consider the Gaussian normalization function that maps all $\{x_i\}$ into corresponding $\{z_i\}$, for $i \in 1 \ldots N$. We define $\mu = \dfrac{1}{N} \sum\limits_{j=1}^{N} x_i$ and $\sigma^2 = \dfrac{1}{N} \sum\limits_{j=1}^{N} (x_i - \mu)^2$ with $\sigma > 0$. The normalized activations $z_i$ are defined as $z_i = \dfrac{x_i - \mu}{\sigma}$. Then, the partial derivatives satisfy*

$$\frac{\partial z_j}{\partial x_i} = \begin{cases} \dfrac{-1 - z_i z_j}{\sigma N} & \text{if } j \neq i \\[3mm] \dfrac{N - 1 - z_i^2}{\sigma N} & \text{if } j = i \end{cases} \tag{20}$$

*Proof.* In deriving $\dfrac{\partial z_j}{\partial x_i}$, we will treat the cases of when $j \neq i$ and when $j = i$ separately. We start by examining intermediate quantities of interest as a matter of convenience for later use. We define helper quantities $u_i = x_i - \mu$. Note that each $u_j$ depends on all of $x_i$ via $\mu$. Next, we write out useful identities

$$\frac{1}{N} \sum_{j=1}^{N} u_j = 0 \tag{21}$$

$$\frac{\partial \mu}{\partial x_i} = \frac{1}{N} \tag{22}$$

$$\frac{\partial u_i}{\partial x_j} = \begin{cases} \dfrac{-1}{N} & \text{if } i \neq j \\[3mm] \dfrac{N-1}{N} & \text{if } i = j \end{cases} \tag{23}$$

We prepare to differentiate with rule of total derivative:

$$\sigma^2 = \frac{1}{N} \sum_{j=1}^{N} u_j^2 \tag{24}$$

$$\frac{\partial(\sigma^2)}{\partial x_i} = \frac{1}{N} \sum_{j=1}^{N} \frac{\partial(\sigma^2)}{\partial u_j} \frac{\partial u_j}{\partial x_i} \tag{25}$$

Making use of equations 21, 22, 23 and 25, We simplify $\dfrac{\partial \sigma}{\partial x_i}$ for any $i$ as follows.

$$\frac{\partial \sigma}{\partial x_i} = \frac{\partial \sigma}{\partial (\sigma^2)} \frac{\partial (\sigma^2)}{\partial x_i} \tag{26}$$

$$= \frac{\partial \sigma}{\partial (\sigma^2)} \frac{1}{N} \sum_{j=1}^{N} \frac{\partial (\sigma^2)}{\partial u_j} \frac{\partial u_j}{\partial x_i} \tag{27}$$

$$= \left[ \frac{1}{2N} (\sigma^2)^{\frac{-1}{2}} \right] \sum_{j=1}^{N} (2u_j) \frac{\partial u_j}{\partial x_i} \tag{28}$$

$$= \frac{1}{2N\sigma} \left[ \left( \sum_{j \neq i}^{N} (2u_j) \frac{\partial u_j}{\partial x_i} \right) + 2u_i \frac{\partial u_i}{\partial x_i} \right] \tag{29}$$

$$= \frac{1}{N\sigma} \left[ \left( \sum_{j \neq i}^{N} (u_j) \frac{-1}{N} \right) + u_i \frac{N-1}{N} \right] \tag{30}$$

$$= \frac{1}{N\sigma} \left[ \left( \underbrace{\sum_{j=1}^{N} (u_j) \frac{-1}{N}}_{=0} \right) + u_i \right] \tag{31}$$

$$= \frac{1}{N\sigma} u_i \tag{32}$$

$$\frac{\partial \sigma}{\partial x_i} = \frac{x_i - \mu}{\sigma N} \tag{33}$$

We apply the quotient rule on $\dfrac{\partial z_j}{\partial x_i}$ when $j \neq i$, then substitute equation 33

$$\frac{\partial z_j}{\partial x_i} = \frac{\partial}{\partial x_i} \left[ \frac{x_j - \mu}{\sigma} \right] \tag{34}$$

$$= \frac{-\sigma \frac{\partial \mu}{\partial x_i} - (x_j - \mu) \frac{\partial \sigma}{\partial x_i}}{\sigma^2} \tag{35}$$

$$= \frac{\frac{-\sigma}{N} - (x_j - \mu) \frac{x_i - \mu}{\sigma N}}{\sigma^2} \tag{36}$$

$$= \frac{-1 - (x_j - \mu) \frac{x_i - \mu}{\sigma^2}}{N\sigma} \tag{37}$$

$$\frac{\partial z_j}{\partial x_i} = \frac{-1 - z_i z_j}{N\sigma} \tag{38}$$

Similarly, when $i = j$,

$$\frac{\partial z_i}{\partial x_i} = \frac{\partial}{\partial x_i} \left[ \frac{x_i - \mu}{\sigma} \right] \tag{39}$$

$$= \frac{\sigma (1 - \frac{\partial \mu}{\partial x_i}) - (x_i - \mu) \frac{\partial \sigma}{\partial x_i}}{\sigma^2} \tag{40}$$

$$= \frac{\sigma - \frac{-\sigma}{N} - (x_i - \mu) \frac{x_i - \mu}{\sigma N}}{\sigma^2} \tag{41}$$

$$= \frac{N - 1 - (x_j - \mu) \frac{x_i - \mu}{\sigma^2}}{N\sigma} \tag{42}$$

$$\frac{\partial z_i}{\partial x_i} = \frac{N - 1 - z_i^2}{N\sigma} \tag{43}$$

$\square$

# B   WEIGHT NORMALIZATION RECAST AS GRADIENT REGRESSION

We show steps to recast the gradient of WN as regression. In WN, A raw weight vector $v$ is normalized and scaled and before being used as coefficient weights. Salimans & Kingma (2016) introduced their transformation as $w = \dfrac{c}{\|v\|}v$, where $c$ is a learned downstream linear scaling parameter. In our regression setup, we ignore $c$. We define $z = \dfrac{v}{\|v\|} = \dfrac{w}{c}$. We derive the analogous relationship between each $\|v\|\dfrac{\partial L}{\partial v_j}$ and the regression on $\left\{\left(z_i, \dfrac{\partial L}{\partial z_i}\right)\right\}_{i=1\ldots N}$.

Note that intercept-0 single variable least squares $\hat{\beta} = \arg\min\limits_{\beta} \|\nabla_z L - \beta z\|^2$ has the solution $\hat{\beta} = \dfrac{(\nabla_z L)^T z}{\|z\|^2} = (\nabla_z L)^T z$, since $\|z\| = 1$.

**Lemma 2.** *Let $v \in \mathbb{R}^N$ be weights, and let $z = \dfrac{v}{\|v\|}$. Let $L$ be a loss function dependent on $z$. Then, for each component $j$, we have*

$$\|v\|\frac{\partial L}{\partial v_j} = \frac{\partial L}{\partial z_j} - \hat{\beta}z_j \tag{44}$$

*where $\hat{\beta} = \arg\min\limits_{\beta} \|\nabla_z L - \beta z\|^2 = (\nabla_z L)^T z$.*

*Proof.* Salimans & Kingma (2016) wrote their gradients as follows:

$$\nabla_c L = \frac{\nabla_w L \cdot v}{\|v\|} \tag{45}$$

$$\nabla_v L = \frac{c}{\|v\|}\nabla_w L - \frac{c\nabla_c L}{\|v\|^2}v \tag{46}$$

In our notation where $z = \dfrac{v}{\|v\|} = \dfrac{w}{c}$, we have

$$\nabla_z L = c\nabla_w L \tag{47}$$

To recover $\hat{\beta}$ We substitute equation 45, and then subsequently equation 47 into equation 46

$$\nabla_v L = \left(\frac{c}{\|v\|}\right)\nabla_w L - \frac{c\nabla_w L \cdot v}{\|v\|^3}v \tag{48}$$

$$\nabla_v L = \frac{1}{\|v\|}\left(c\nabla_w L - \frac{(c\nabla_w L \cdot v)v}{\|v\|^2}\right) \tag{49}$$

$$\|v\|\nabla_v L = \nabla_z L - \frac{(\nabla_z L \cdot v)v}{\|v\|^2} \tag{50}$$

$$\|v\|\nabla_v L = \nabla_z L - \underbrace{(\nabla_z L \cdot z)}_{\hat{\beta}}z \tag{51}$$

The result follows: for loss $L$ and for each component $j$, we have

$$\|v\|\frac{\partial L}{\partial v_j} = \frac{\partial L}{\partial z_j} - \hat{\beta}z_j \tag{52}$$

$\square$

# C   ADDRESSING SMALL BATCHES WITH LEAST-SQUARES REGULARIZATION

Let $b$ be an index for different batches; let $X$ refer to data inputs into the neural network (for example image and class label) within a single step of training, and let $X^{(b)}$ refers to the value of all data

Table 2: Streaming regularization is less affected by small batch sizes (ResNet-34-v2, batch size 2)

| Normalization | CIFAR-10 Accuracy | CIFAR-10 Cross Entropy |
|---|---|---|
| Our Best Hyperparameter | **0.9091** | **0.3627** |
| Our Worst Hyperparameter | 0.9005 | 0.4086 |
| BN (Ioffe & Szegedy, 2015) | 0.8903 | 0.4624 |
| Renorm (Ioffe, 2017) | 0.9033 | 0.3823 |
| Identity | 0.9086 | 0.6934 (0.4229 at best point) |

inputs in batch $b$. In our work, we keep track of am exponential running estimates across batches, $\hat{\alpha}^* \sim \mathbb{E}_b \mathbb{E}_i \left[ \left. \frac{\partial L}{\partial z_i} \right|_{X = X^{(b)}} \right]$ and $\hat{\beta}^* \sim \mathbb{E}_b \mathbb{E}_i \left[ \left. z_i \frac{\partial L}{\partial z_i} \right|_{X = X^{(b)}} \right]$ that marginalize the $(B, H, W)$ dimensions into accumulators of shape $C$. The $b$ subscript of the outer expectation is slightly abusive notation indicating that $\hat{\alpha}^*$ and $\hat{\beta}^*$ are running averages across recent batches with momentum as a hyperparameter that determines the weighting. We regularize the gradient regression with virtual activations and virtual gradients, defined as follows. We append two virtual batch items, broadcast to an appropriate shape, $x_+ = \mu_b + \sigma_b$ and $x_- = \mu_b - \sigma_b$. Here, $\mu_b$ and $\sigma_b$ are batch statistics of the real activations. The concatenated tensor undergoes standard BN, which outputs the usual $\{z_i\}$ for the real activations, but $z_+ = 1$ and $z_- = -1$ for the virtual items. The $z_+$ and $z_-$ do not affect the feed forward calculations, but they receive virtual gradients during back-propagation:

$$\frac{\partial L}{\partial z_+} = \hat{\alpha}^* + \hat{\beta}^* \tag{53}$$

$$\frac{\partial L}{\partial z_-} = \hat{\alpha}^* - \hat{\beta}^* \tag{54}$$

Virtual data $\left( z_+, \frac{\partial L}{\partial z_+} \right)$ and $\left( z_-, \frac{\partial L}{\partial z_-} \right)$ regularizes the *gradient-least-squares* regression. $\frac{\partial L}{\partial z_+}$ and $\frac{\partial L}{\partial z_-}$ eventually modify the gradients received by the real $x_i$ activations. The virtual data can be weighted with hyperparameters. In our experiments, we see improvements, robust to a hyperparameter cross-product search over the weightings and the momentum for $\hat{\alpha}^*$ and $\hat{\beta}^*$. The momentum for $\hat{\alpha}^*$ and $\hat{\beta}^*$ were in $\{.997, .5\}$ and the virtual item weights were in $\{2^{i-1}\}_{i \in \{0,1,2,3\}}$. The performance of larger batches are not recovered; regularized regression could not be reasonably expected to recover the performance of regressing with more data. See table 2 for final validation performances with a reference Tensorflow ResNet-34-v2 implementation on batch size of two. The baseline evaluation with identity (no normalization) experienced noticeable overfitting in terms of cross entropy but not accuracy. The base learning rate was multiplied by $\frac{1}{64}$ relative to the baseline rate used in runs with batch size 128.

