# OpenReview forum: "Normalization Gradients are Least-squares Residuals"
_ICLR.cc/2019/Conference_

### Official Review · AnonReviewer1 · 2018-11-02
**A premature paper proposing a novel interpretation of Batch Normalisation**

**Rating:** 3
**Confidence:** 4

**Review:**

The paper aims at a better understanding of the positive impacts of Batch Normalisation (BN) on network generalisation (mainly) and  convergence of learning. First, the authors propose a novel interpretation of the BN re-parametrisation. They show that an affine transform of the variables with their local variance (scale) and mean (shift) can be interpreted as a decomposition of the gradient of the objective function into a regressor assuming that the gradient is parallel to the variables (up to a shift) and the residual part which is the gradient w.r.t. to the new variables. In the second part of the paper, authors review various normalisation proposals (differing mainly in the subset of variables over which the normalisation statistics is computed) as well as the known empirical findings about the dependence of BN on the batch size. The paper presents an experiment that combines two normalisation variants. A further experiment strives at regularising BN for small batch sizes.

Unfortunately, it remains unclear what questions precisely the authors answer in the second part of the paper and, what is more important, how they are related to the novel interpretation of BN presented in the first part. This interpretation holds for any function and can be possibly seen as a gradient pre-conditioning. However, the authors do not "extend" it towards the gradients w.r.t. the network parameters and do not consider the specifics of the learning objectives (a sum of functions, each one depending on one training example only). The main presented experiment combines layer normalisation with standard batch normalisation for a convolutional network. The first one normalises using the statistics over channel and spatial dimensions, whereas the second one uses the statics over the batch and spatial dimensions. The improvements are rather marginal, but, what is more important, the authors do not explain how and why this proposal follows from their new interpretation of BN.

Overall, in my view, this paper is premature and not appropriate for publishing at ICLR in its present form.

---

> ### Author Response · Authors · 2018-11-26
> **We have made notable adjustments to our language in the abstract, the TLDR, and the main body of the text, in light of your review.**
>
> Dear Paper923 AnonReviewer3,
> Thank you for your criticisms.  We have made notable adjustments to our language in the abstract, the TLDR, and the main body of the text, in light of your review. Regarding concerns related to ignoring the affine transform downstream of the gaussian normalization, we have rephrased the text to emphasize that it is done without loss of generality. This is WLOG in the sense that the affine transform after gaussian normalization can be absorbed into the rest of the network. Also, we would like to emphasize that we think of division by the standard deviation during training as a non-affine transform. One way to make that division affine is to use the BN running-variance instead of the batch variance during training -- but this alternative generally known (but not well stated in literature) to lead to poor performance.

---

### Official Review · AnonReviewer2 · 2018-11-03
**It has not been sufficiently demonstrated that the new perspective regarding batch normalization presented in this work is actually useful for either improving or explaining BN.**

**Rating:** 4
**Confidence:** 4

**Review:**

The primary technical contribution comes from Section 2, where it is demonstrated that the normalized back-propagated gradients obtained from a BN layer can be viewed as the residuals of the gradients obtained without BN regressed via a simple two-parameter model of the activations.  In some sense though this result is to be expected, since centering data (i.e., removing the mean as in BN) can be generically viewed as computing the residuals after a least squares fit of a single constant, and similarly for de-trending with respect to a single independent variable, in this case the activations.  So I'm not sure that Theorem 1 is really that much of an insightful breakthrough, even if it may be nice to work through the precise details in the specific case of a BN layer and the relationship to gradients.

But beyond this a larger issue is as follows:  This paper is framed as taking a step in explaining why batch normalization (BN) works so well.  For example, even the abstract mentions this as an unsettled issue in motivating the proposed analysis.  However, to me the interpretation of BN as introducing a form of least squares fit does not really extend our understanding of why it actually works better in practice, and this is the biggest disconnect of the paper.  The new perspective presented might be another way to interpret BN layers, but it unfortunately remains mostly unanswered exactly why this new perspective is relevant in actually explaining BN behavior.

The presented normalization theory is also used to motivate heuristic modifications to standard BN schemes.  For example, the paper proposed concatenating BN with a layer normalization layer, demonstrating some modest improvement on CIFAR-10 data.  But again, I don't see how viewing these normalization schemes as least-squares residuals motivates such concatenation any more than the merits of the original versions themselves.  Moreover, it is not even clear that BN+LN is in fact generally better since only a single data set is considered.  There are also no comparisons against competing BN modifications such as switch normalization (Luo et al. 2018) which also involves a hybrid method combining aspects of LN and BN.  Why not compare against approaches like this?

To conclude, in Section 6 the paper asks "Why do empirical improvements in neural networks with BN keep the gradient-least-squares residuals and drop the explained portion?"  But this question is not at all answered but rather deferred to future work.  For me this was a disappointment as this would seem to be an essential ingredient for actually developing a meaningful theory for why BN is helpful in practice.


Other comments:

* The analysis from Section 2, including Theorem 1, assume that the BN parameters c and b can be ignored (presumably this means fixing c = 1 and b = 0).  I did not carefully check the details, but do all the same derivations and conclusions still seamlessly go through when these parameters have general values that deviate from this standard initialization?  If not, then I don't really see what is the practical relevance, since once learning begins, both b and c will typically shift to arbitrary values.  Below eq. (1) it states that c and b are only ignored for clarity, but then later I did not see any subsequent discussion to handle the general case, which is what would be actually needed for explaining BN behavior in practice.

* Please run a speck-checker.  Example, "On some leve, the matrix gradient ..."

* The paper cites (Lipton and Steinhardt, 2018) in arguing that reasons for the effectiveness of BN are lacking.  Indeed (Lipton and Steinhardt, 2018) criticize the original BN paper for conflating speculation with explanation, or more precisely, framing speculation about why BN should be helpful as an actual true explanation without clear evidence.  But to me this submission is hovering somewhere in the same category, speculating that regressing away certain portions of the gradient could be useful but never really providing concrete evidence for why this should offer an improvement.

---

> ### Author Response · Authors · 2018-11-26
> **We have dialed back our language in the abstract, the TLDR, and the main body of the text to reflect your perspective on our work.**
>
> Dear Paper923 AnonReviewer2
> Thank you for your criticisms.  We have dialed back our language in the abstract, the TLDR, and the main body of the text to reflect your perspective on our work. We apologize for the typos in the earlier version, and we have been more diligent in this update. Also, we have clarified some of the language around the downstream affine transformation. Ignoring the affine transform is done without loss of generality, in the sense that they can be absorbed into the rest of the network without impacting our view of the gradients of the gaussian normalization. Our experiments were meant to be toy-examples of how one might better understand what happens to the gradient regression under adjustments to BN; ideally, we would like to design a new normalization that outperforms switch normalization, but we have not been able to do that here.

---

### Official Review · AnonReviewer3 · 2018-11-07
**The discussion and conclusion drawn from the experimental part are quite arguable. In my opinion, the paper would gain much more impact if the view developed in this paper were illustrated by more convincing experiments.**

**Rating:** 4
**Confidence:** 5

**Review:**

The authors propose a new interpretation of the batch normalization step inside a neural network.
The main result shows that the backpropagation of the gradient of some loss function through a batch normalization can be seen as a scaled residual of a least square linear fit. This new interpretation is extended to other normalization technics used in the literature and thus give a "unified" view of such methods.

The idea is simple yet very interesting and well introduced. The theoretical results are good and the proofs are well written and easy to follow.

However the arguments brought forward by this new vision of batch normalization in applications look light (see sections 3.3, 4.1, 4.2). A more detailed interpretation of this new vision on a single application and its impact would have been preferred than numerous applications as it is done in this paper.
Not all the existing normalization methods have been extended with success yet, this makes this unified vision a bit less convincing.

---

> ### Author Response · Authors · 2018-11-26
> **Thank you for your thoughtful criticism, and especially for your kind comments.**
>
> Dear Paper923 AnonReviewer3,
> Thank you for your thoughtful criticism, and especially for your kind comments. We have toned down our language broadly in our revision, and removed all mentions of a "unified view." We agree deeply with your note on needing more focus in the experiments; we would have liked to followup with a cleaner, more convincing application. Sadly, we could not deliver this under the resource constraints.

---

### Meta-Review · Area_Chair1 · 2018-12-06
**a new interpretation of batch norm, but not clear what we gain from it**

**Confidence:** 5
**Recommendation:** Reject

**Metareview:**

This paper interprets batch norm in terms of normalizing the backpropagated gradients. All of the reviewers believe this interpretation is novel and potentially interesting, but that the paper doesn't make the case that this helps explain batch norm, or provide useful insights into how to improve it. The authors have responded to the original set of reviews by toning down some of the claims in the original paper, but haven't addressed the reviewers' more substantive concerns. There may potentially be interesting ideas here, but I don't think it's ready for publication at ICLR.